# Response of Phenolic Compounds in *Lippia graveolens* Kunth Irrigated with Aquaculture Wastewater and Steiner Solution

María Isabel Nieto-Ramírez [1] , Ana Angélica Feregrino-Pérez [2], Humberto Aguirre Becerra [2] , Benito Parra-Pacheco [1], Mónica Vanessa Oviedo-Olvera [1] and Juan Fernando García-Trejo [2,*]

1   División de Investigación y Posgrado, Facultad de Ingeniería, Universidad Autónoma de Querétaro, Carretera a Chichimequillas Km. 1 s/n Amazcala, El Marques 76265, Querétaro, Mexico
2   Cuerpo Académico de Bioingeniería Básica y Aplicada, Facultad de Ingeniería, Universidad Autónoma de Querétaro, Cerro de las Campanas s/n, Las Campanas, Santiago de Querétaro 76010, Querétaro, Mexico
*   Correspondence: fernando.garcia@uaq.mx

**Abstract:** *Lippia graveolens* is one of the most important aromatic species in Mexico due to antioxidant and antibiotic activities reported in its essential oil. The aim of this work was to assess the effect of irrigation with aquaculture wastewater and salicylic acid addition on the production of phenolic compounds in *L. graveolens*. *L. graveolens* plants (14) were irrigated with aquaculture wastewater and (14) using Steiner solution for 28 days; at the same time, salicylic acid was exogenously applied at 0.0 (control), 0.5 and 1.0 mM concentrations in both treatments at 5 and 19 experimental days. The total phenolic content was measured by Folin–Ciocalteu, the flavonoid content was determined by the aluminum chloride method, and the antioxidant capacity was measured by DPPH and FRAP assays. The results showed an increase in the total phenolic and flavonoid content in plants irrigated with aquaculture wastewater solution ($17.25 \pm 2.35$ to $38.16 \pm 4.47$ mg eq $GA \cdot g^{-1}$ W). The antioxidant capacity was higher in plants irrigated with Steiner solution (98.52 mg eq $T \cdot g^{-1}$ W). In conclusion, *L. graveolens* irrigated with aquaculture wastewater leads to an increase in the total phenolic content and Steiner-solution antioxidant capacity in plants.

**Keywords:** antioxidant capacity; cultivation conditions; elicitor; oregano; secondary metabolites

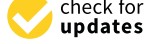



## 1. Introduction

*Lippia graveolens* Kunth (*L. graveolens* Kunth), commonly known as Mexican oregano, is one of the most important aromatic plants in Mexico. Antioxidant and antibiotic activities have been reported in its essential oil [1]. Despite its high demand, it is obtained from wild populations with a very irregular harvest and with a variable quality [2]. The quality in the aromatic species is based on the content of volatile compounds in its essential oil, such as phenols, terpenes and sesquiterpenes [3]. Nevertheless, the phenolic compound content depends on the cultivation area, genetic material, and climatic conditions [4]. According to Calvo–Irabién [5], edaphoclimatic and cultivation conditions modify the content of carvacrol, thymol, and sesquiterpenes in *L. graveolens* essential oil. Additionally, the high genetic variation shown in Mexican oregano can modify the secondary metabolites' content, though this variation was low among gene diversity and chemo-type [6]. Additionally, the macro- and microclimate conditions modify the phenolic monoterpene content in *L. graveolens* growing in the wild [7].

Different strategies have been used to enhance the total phenolic content, such as abiotic elicitation (e.g., light [8] and nutrient deficit [9,10]) and biotic elicitation (e.g., jasmonic acid and salicylic acid [11]). According to Bueno–Duran [12], different spectra of light affect the monoterpene content; based on this, red light increases the carvacrol content, blue light increases the thymol content in *L. graveolens*, and UV-C light application increases the total phenolic content [13]. Moreover, nitrogen deficiency enhances secondary metabolites' production, such as phenolic compounds and flavonoids in Greek oregano [14].

Wastewater from aquaculture contains different compounds, such as ammonia, which need to be removed every week to improve fish growth, and aquaponics activity can use these waters for feeding and growing aromatic plants [15]. According to [16], the total phenolic content in *Ocimun basilicum* increased in an aquaponics system culture. Additionally [17,18], fish-effluent irrigation, used as aquaculture wastewater in *Origanum syriacum* L., and *Origanum mejorana* L., increased the essential oil yield. On the other hand, Steiner solution and other nitrogen fertilizations have been used to evaluate the effect on biomass production and essential oil quality. According to Azizi [19], nitrogen fertilization (250 ppm) in tree oregano (*Origanum vulgare* L.) populations increase the dry-matter production but decrease the essential oil content. However, specific secondary metabolites, such as carvacrol, γ-terpinene and p-cimene, were unaffected. Furthermore, different levels of nitrogen fertilization were applied in *Origanum vulgare* ssp. Hirtium (Link) letswaart to evaluate the in-vitro bioaccessibility and activity of compounds. In this study, unfertilized plants showed the highest total phenolic content and antioxidant activity [14].

Salicylic acid (SA) is a phytohormone that regulates the biosynthesis of different secondary metabolites (e.g., anthocyanins, alkaloids, glucosinolates, and phenolics) [20]. Different studies applied SA at different concentrations to improve the total phenolic content in *Mentha piperita* (0.5, 1, 2 and 3 mM SA) [21,22], *Petroselinum crispum* (50 μM as well as 1 and 2 mM SA) [23], *Majorana hortensis* (0.01, 0.1 and 1.0 mM SA) [24] and *Thymbra spicata* (2.5 and 5.0 mM SA) [25] under different cultivation conditions. On the other hand, repeats of SA applications (7, 14, and 21 days after foliar application) can improve secondary metabolites' production in strawberries [26], *Ajuga integrifolia* (21 days) [27], and *Mentha piperita* (14 days) [28]. However, no studies on elicitation with SA and irrigation with aquaculture wastewater in *L. graveolens* have been reported. This study aimed to assess the effect of aquaculture effluents' irrigation and SA elicitation on phenolic compounds in *L. graveolens*.

## 2. Materials and Methods

### 2.1. Plant Material and Culture Conditions

The experiment was carried out in a Gothic-style greenhouse. Fourteen *L. graveolens* plants were obtained from a local plant shop in La Higuera, Cadereyta, Querétaro with a special management license and were identified by the QMEX botanical department of the Autonomous University of Querétaro. Each plant was transplanted in plastic bags with tezontle and peat moss and was acclimatized inside the greenhouse for five days at 22 °C, 52% relative humidity and an irradiance of 300 μmol m$^{-2}$ s$^{-1}$ for a 16/8 h (day/night) period. The age and initial culture conditions were unknown.

### 2.2. Irrigation

Two solutions were used for irrigation: (1) wastewater from a juvenile fish culture (25–50 g) that contained organic matter (8.5–758 mg·L$^{-1}$ COD), dissolved solids (650 mg·L$^{-1}$ TDS) and nutrients such as nitrogen (0.9 ± 0.45 mg·L$^{-1}$ NH$_3$-N, 1.28 ± 0.99 mg·L$^{-1}$ NO$_2^-$, 13.73 ± 15.28 mg·L$^{-1}$ NO$_3^-$) and phosphorus (5.2–5.4 mg·L$^{-1}$ PO$_4^{3-}$); and (2) a Steiner solution with essential chemical compounds, such as nitrogen, phosphorus, potassium, calcium, magnesium and other elements, in their ionic form [28]. A total of 1000 L of this solution was prepared with 479 g of KNO$_3$, 170 g of NH$_4$H$_2$PO$_4$, 476 g of Mg(NO$_3$)$^2$·6H$_2$O, 936 g of Ca(NO$_3$)$^2$, 89.6 mL of H$_2$SO$_4$, 422 g of Mg EDTA, 3.82 g of H$_3$BO$_3$, 7.53 g of Mn EDTA, 0.714 g Zn EDTA and 7.53 g of Mn EDTA.

### 2.3. Elicitation

Two SA concentrations (0.5 and 1.0 mM) were used as foliar spray two times, and one with water was used as a control (0.0 mM) (Figure 1). These salicylic acid concentrations are within the concentration range applied in aromatic plants [21,23].

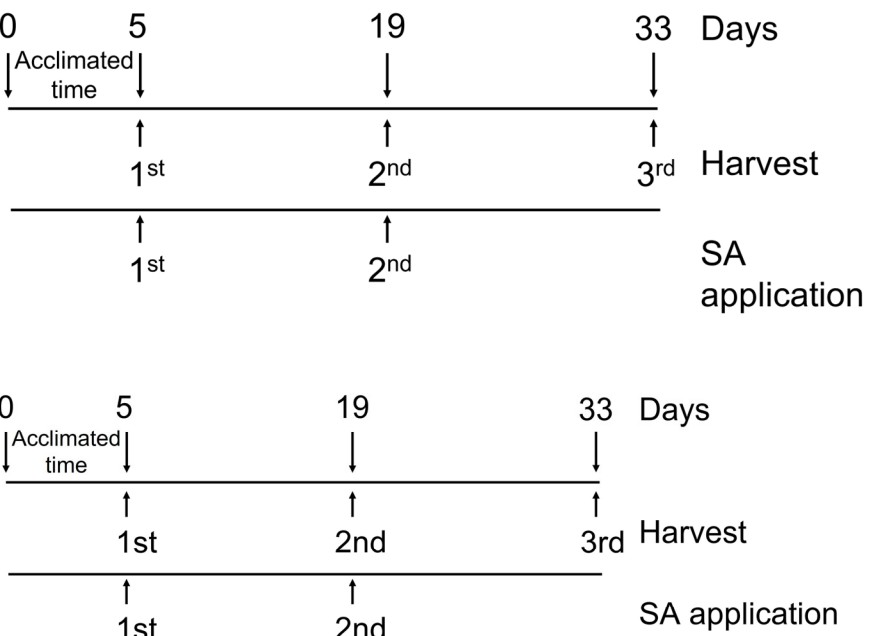

**Figure 1.** Timeline of experiment for harvest and SA applications.

### 2.4. N and P Determination

Nitrites ($NO_2^-$), nitrates ($NO_3^-$), and phosphates ($PO_4^{3-}$) were measured by spectrophotometric methods every week. Nitrites were measured by the diazotization method (HACH Method 8507, 2010), nitrates by the cadmium reduction method (HACH Method 8171, 2010), and phosphates by the ascorbic acid method (HACH Method 8048, 2010).

### 2.5. Vegetative Sample

Plant samples were obtained at three different times (0, 14, and 28 experimental days) and were dehydrated at 28 °C during three days in paper bags to protect them from direct light. Dried samples were ground on a blade mill and sieved (no. 40) and were stored in amber flasks protected from light until use.

### 2.6. Phenolic Compounds Extraction

The extraction of phenolic compounds was carried out with 100 mg of each previously processed sample. One milliliter of a methanol:water:formic acid solution (80:18:2) was added, and the mixture was sonicated for 30 min. After the extraction time, each sample was centrifuged at $10,000\times g$ rpm in a METRIX Dynamica® (Dynamica a techcomp Company, Livingston, United Kingdom) centrifuge at 4 °C for 15 min. The supernatant was stored in a new tube, and a second extraction was performed. Extracts were stored at 4 °C until use.

### 2.7. Phenolic Content Determination

Phenolic compounds' content was carried out by the Folin–Ciocalteu method described by [29], adapted to microplate reader spectrometry (Multiskan Go®Thermo Fisher Scientific Oy, Finland). A sodium carbonate solution was prepared at 7.5%, and a 0.2 N Folin–Ciocalteu reagent was used at a 1:10 dilution. A total of 30 μL of extract and 120 μL of carbonate solution were placed in the microplate wells, and 150 μL of Folin–Ciocalteu reagent was added. Different concentrations (0.02, 0.04, 0.08, 0.10, 0.11, 0.12 mg·mL$^{-1}$) of gallic acid were used for curve calibration, and the results were expressed as mg gallic acid equivalents per g of dry weight (mg GAE·g$^{-1}$ DW).

### 2.8. Flavonoid Content Determination

Flavonoids were determined by the aluminum chloride method described by [30]. A 10% aluminum chloride, 5% sodium nitrite, and 1 N sodium hydroxide solution was

prepared. A total of 40 µL of distilled water, 100 µL of extract and 30 µL of sodium nitrite (5% $w/v$) was combined into microtubes. After 5 min, 30 µL of sodium chloride was added, and 200 µL of sodium hydroxide was incorporated after 1 min. Finally, 240 µL of distilled water was added, and 300 µL was taken from each microtube and placed in microplate wells. Different concentrations (0.006, 0.009, 0.018, 0.036, 0.15, 0.3, and 0.6 mg·mL$^{-1}$) of catechin hydrate were used for the calibration curve, and the results were expressed as mg catechin hydrate equivalents per g of dry weight (mg CHE·g$^{-1}$ DW).

### 2.9. Antioxidant Capacity

2.9.1. DPPH Assay

The antioxidant capacity by the 2,2-Diphenyl-1-picrylhydrazyl (DPPH) assay was carried out according to [31]. DPPH radical (0.15 mM) was prepared in methanol, and the absorbance was measured at 480 nm using a Multiskan Go® microplate reader. Briefly, 20 µL of extract and 280 µL (0.15 mM) of DPPH solution in methanol were incubated in the dark for 30 min. Trolox in different concentrations (0.05, 0.08, 0.11, 0.14, 0.17, and 0.2 mg·mL$^{-1}$) were used for the calibration curve. The results were expressed as mg Trolox equivalents per g of dry weight (mg T·g$^{-1}$ DW).

2.9.2. FRAP Assay

A ferric reducing antioxidant power (FRAP) test was determined by the method described by [32]. The FRAP reagent was prepared by mixing 50 mL of 300 mM acetate buffer (pH 3.6) and 5 mL of 10 mM 2,4,6-tripyridyl-2-triazine (TPTZ) in 40 mM HCl and 5 mL of 20 mM $FeCl_3$. Aliquots containing 280 µL of the reagent were taken and placed in a 96-well plate with 20 µL of the extract. The reaction was carried out for 30 min at room temperature and protected from light. Readings were taken at 630 nm with a microplate spectrophotometer. The results were expressed as mg Trolox equivalents per g of dry weight (mg T·g$^{-1}$ DW).

### 2.10. Statistical Analysis

Data were analyzed by a normality test to obtain the type of distribution using Statgraphics Centurion XV, Virginia, E.U software, version 15.2.06. As a result, kurtosis data were −0.99, showing a normal distribution with a leptokurtic form. Furthermore, a one-way ANOVA was performed for irrigation and elicitation treatments. Furthermore, a multifactorial ANOVA was performed to statistical significance between the type of irrigation and SA concentrations. Statistical significance was considered with a probability value of $p < 0.05$. All experiments were carried out in triplicate, and the results were expressed as the mean ± standard error (SE).

## 3. Results

### 3.1. Water Quality (N and P Determination)

The results showed a high nitrate, nitrite, and phosphate content in the Steiner solution (Table 1). The differences in these contents affect the plant nutrition and could impact the secondary metabolites' production.

**Table 1.** Aquaculture wastewater and Steiner-solution water quality for nitrates ($NO_3^-$), nitrites ($NO_2^-$), and phosphates ($PO_4^{3-}$) (mg/L).

| | Week 1 | | Week 2 | | Week 3 | | Week 4 | |
|---|---|---|---|---|---|---|---|---|
| | SS [1] | AQW [2] | SS | AQW | SS | AQW | SS | AQW |
| $NO_3^-$ | 127.3 ± 4.24 | 108.2 ± 2.22 | 207.7 ± 1.23 | 139.2 ± 6.07 | 290.3 ± 4.32 | 122.3 ± 3.37 | 235.4 ± 2.15 | 103.8 ± 4.57 |
| $NO_2^-$ | 1.361 ± 0.04 | 0.159 ± 0.07 | 0.313 ± 0.04 | 0.403 ± 0.06 | 3.176 ± 0.28 | 0.562 ± 0.08 | 0.515 ± 0.02 | 0.163 ± 0.02 |
| $PO_4^{3-}$ | 17.79 ± 0.14 | 24.16 ± 0.27 | 43.09 ± 0.35 | 9.75 ± 1.03 | 114.9 ± 3.18 | 12.52 ± 1.64 | 102.2 ± 6.41 | 10.97 ± 2.06 |

[1] SS: Steiner solution, [2] AQW: Aquaculture wastewater.

### 3.2. Phenolic Content

The phenolic compound content in *L. graveolens* irrigated with aquaculture wastewater and elicited by salicylic acid was analyzed by multifactorial ANOVA analysis. According to the results, any of the factors (salicylic acid concentrations) showed *p*-values under 0.05; therefore, there are no statistical differences in the Mexican oregano elicited at different salicylic acid concentration. However, the highest phenolic compound content at harvest day 14 occurred in plants irrigated with aquaculture wastewater ($35.47 \pm 3.74$ mg eq $GA \cdot g^{-1}$ W), being higher than in those irrigated with Steiner solution ($24.29 \pm 6.51$ mg eq $GA \cdot g^{-1}$ W) (Figure 2). On the other hand, at harvest day 28, the phenolic compound content decreased in plants irrigated with both irrigations.

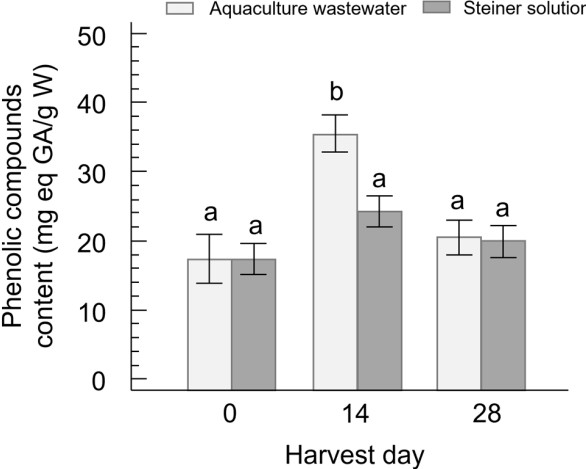

**Figure 2.** Total phenolic content in *L. graveolens* irrigated with aquaculture wastewater and Steiner solution at three different harvest days (0, 14, and 28). Data were evaluated via one-way ANOVA separately for each irrigation: aquaculture wastewater and Steiner solution. Different letters indicate significant difference among different harvest days at $p \leq 0.05$.

### 3.3. Flavonoid Content

*L. graveolens* plants elicited with salicylic acid at different concentrations showed no statistical differences ($p < 0.05$) for the flavonoid content. Additionally, plants irrigated with aquaculture wastewater and Steiner solution did not show statistical differences in terms of the flavonoid content. However, an increase in these compounds was observed in plants irrigated at harvest day 14 and was maintained at harvest day 28 (Figure 3).

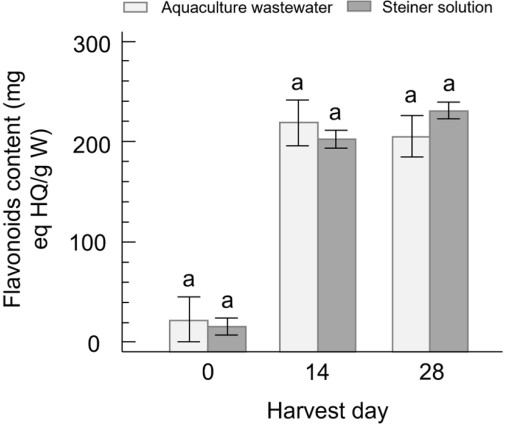

**Figure 3.** Flavonoid content in *L. graveolens* irrigated with aquaculture wastewater and Steiner solution at different harvest days (0, 14, and 28). Data were evaluated via one-way ANOVA separately for each irrigation. Different letters indicate significant difference among different harvest days at $p \leq 0.05$.

*3.4. Antioxidant Activity*

3.4.1. DPPH Assay

The DPPH radical is a stable molecule that is mainly used to determine the antioxidant capacity via its hydrogen acceptor capability in relation to antioxidants. *L. graveolens* with Steiner-solution irrigation showed the highest antioxidant capacity by DPPH assay with statistical differences ($p < 0.05$) when compared to plants irrigated with aquaculture wastewater (Figure 4). On the other hand, this antioxidant capacity was similar at harvest days 14 and 28. The antioxidant capacity in plants irrigated with aquaculture wastewater did not show statistical significances at harvest days 14 and 28.

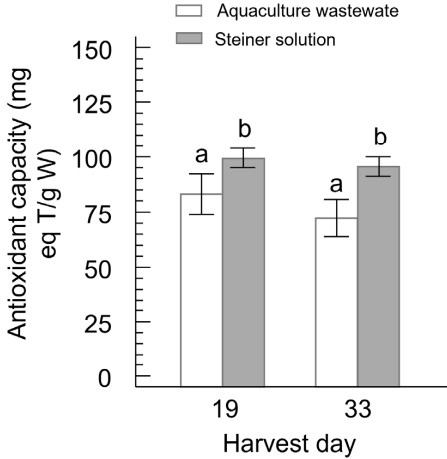

**Figure 4.** Antioxidant capacity by DPPH assay in *L. graveolens* irrigated with aquaculture wastewater and Steiner solution at two different harvest days (14 and 28). Data were evaluated via one-way ANOVA separately for each irrigation. Different letters indicate significant difference among different harvest days at $p \leq 0.05$.

3.4.2. FRAP Assay

Mexican oregano irrigated with Steiner solution showed the highest antioxidant capacity by FRAP assay with statistical differences ($p < 0.5$) when compared to plants irrigated with aquaculture wastewater (Figure 5). Plants irrigated with aquaculture wastewater and Steiner solution increased their antioxidant capacity at harvest day 28. At harvest day 14, the antioxidant capacity was not different between plants irrigated with either treatment.

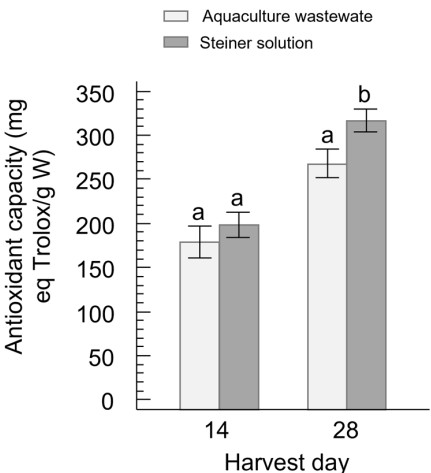

**Figure 5.** Antioxidant capacity by FRAP assay in *L. graveolens* irrigated with aquaculture wastewater and Steiner solution at two different harvest days (14 and 28). Data were evaluated via one-way ANOVA separately for each irrigation. Different letters indicate significant difference among different harvest days at $p \leq 0.05$.

## 4. Discussion

The phenolic compound content was higher in *L. graveolens* irrigated with aquaculture wastewater. This result was similar to that obtained in *Origanum vulgare* L. ssp. Hirtum (Link) Ietswaart fertilized at different nitrogen levels (0–150 kg of N ha$^{-1}$), which showed the highest phenolic content in unfertilized plants, with an increasing nitrogen level leading to a decreasing phenolic content [33]. Additionally, the higher phenolic content in *Moringa oleifera* was at a low nitrogen level fertilization (0.7 mg·L$^{-1}$ of N-NO$_3$) [14]. In contrast, wheat plants fertilized with different nitrogen concentrations (0, 75, 150, and 195 kg N ha$^{-1}$) did not present statistical differences in terms of the phenolic content [34]. These differences could be explained by nitrogen limitation in aquaculture wastewater. According to [35], aquaculture wastewater showed a lower nitrogen concentration than nutritive solution. Nutrient anions, such as nitrates (NO$_3{}^-$), nitrites (NO$_2{}^-$), and phosphates (PO$_4{}^{3-}$), are the principal nutrients that could be found in aquaculture wastewater. These nutrients at specific concentrations levels could become toxic to fish. As a result, 10% of the total wastewater should be replaced with fresh water. This wastewater can be used in plant growth as a wastewater treatment. Nitrogen is involved in the vegetative growth of plants via protein and carbohydrate synthesis; however, nitrogen deficiency stimulates an increase in secondary metabolites. This stimulation occurs via the carbon–nitrogen relation in plants: when nitrogen decreases, carbon increases, promoting the synthesis of carbon-based compounds such as phenolic compounds [36].

The total phenolic content in *L. graveolens* did not present a statistical significance with the application of SA. These results were different in *Mentha piperita* [21], *Petroselinum cripum* L. [23] and *Thymbra spicata* [25] elicited with SA, which showed an increase in the total phenolic content. Another study reported a decrease in the phenolic content in *Eruca veiscari* subs. Sativa elicited with 100 ppm [37]. According to these studies, it is necessary to assess different concentrations of salicylic acid to evaluate the phenolic content in *L. graveolens*.

The flavonoid content was not different in *L. graveolens* elicited with SA and irrigated with aquaculture wastewater and Steiner solution. However, these compounds increased at 14 and 28 harvest days. These results are different to those reported in *Calendula officinalis* fertilized at a low nitrogen concentration [38]. On the other hand, the flavonoid content in *Cyclocarya paliurus* fertilized with an intermediate nitrogen level (3.4 g/plant NH$_4$NO$_3$) was higher than when using low and high nitrogen levels [39]. This result was similar to that obtained in *L. graveolens*, where an increase in the flavonoid content was shown at 28 harvest days in plants irrigated with Steiner solution. The relation between the flavonoid production and nitrogen level could be explained by the bioavailability of nitrogen and its absorption of the species. Furthermore, these differences could be explained by the activation of the phenylalanine ammonia lyase (PAL) enzyme resulting from nitrogen deficiency [40].

The DPPH assay determines the ability of antioxidants to bind free radicals, and the FRAP assay determines the reducing power [41]. In this study, the antioxidant capacity via DPPH and FRAP was higher in *L. graveolens* irrigated with Steiner solution. These results are similar to those obtained in colored potatoes (*Solanum tuberosum* L. subsp. Andigenum) fertilized with different nitrogen levels. The results of this study showed an increase in the antioxidant capacity, measured by FRAP, with an increased nitrogen level [42]. In contrast, oregano (*Origanum vulgare*) fertilized with organic fertilizer showed a higher antioxidant capacity than that with mineral fertilizer [43]. Additionally, lavender flowers showed the same trend of a high antioxidant capacity with a low nitrogen concentration when compared to those fertilized at high nitrogen concentrations [44]. These results could be explained by the relationship between polyphenols and the structural chemistry for free radical-scavenging activities [45]. Additionally, antioxidant capacity depends on specific polyphenols, such as phenols, flavonoids, flavones, catechins, and others [46]. Querétaro belongs to a semi-desert region with environmental conditions that are hostile to its flora and fauna. *Lippia graveolens*, a plant in this area, has a small size and leaves but a high phenolic content associated with

environmental stress [5]. Due to the brief experimental duration of this study, foliar and production data were not registered.

Future studies must be conducted to improve secondary metabolites in *L. graveolens* Kunth irrigated with aquaculture wastewater containing different species of fish with different ages and feeds. Additionally, Mexican oregano culture conditions must be established in order to improve specific secondary metabolites such as carvacrol and thymol.

## 5. Conclusions

The present study was designed to assess the effect of aquaculture-effluent irrigation and salicylic-acid elicitation on phenolic compounds in *L. graveolens*. Aquaculture wastewater increases the phenolic content in *L. graveolens* plants from the Querétaro semi-desert under stress conditions, and the Steiner solution enhances the antioxidant capacity via a DPPH assay. In relation to *L. graveolens* from the Querétaro semi-desert, more studies are needed on the effect of environmental conditions, such as the temperature variation, light intensity, and soil–plant relationships, on the secondary metabolite content and aquaculture wastewater irrigation effect in order to enhance the flavonoid content and antioxidant capacity.

**Author Contributions:** Conceptualization, J.F.G.-T.; Formal analysis, M.V.O.-O.; Funding acquisition, J.F.G.-T.; Investigation, M.I.N.-R., B.P.-P. and M.V.O.-O.; Methodology, A.A.F.-P.; Software, H.A.B.; Supervision, A.A.F.-P.; Validation, H.A.B. All authors have read and agreed to the published version of the manuscript.

**Funding:** This research received no external funding.

**Institutional Review Board Statement:** Not applicable for studies not involving humans or animals.

**Informed Consent Statement:** Not applicable for studies.

**Data Availability Statement:** Data can be found in Institutional repository.

**Acknowledgments:** The authors are thankful to CONACyT (Consejo Nacional de Ciencia y Tenología) for scholarship support (653909).

**Conflicts of Interest:** The authors declare no conflict of interest.

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
