# Peer review of "Response of Phenolic Compounds in Lippia graveolens Kunth Irrigated with Aquaculture Wastewater and Steiner Solution"

_2037-0164, doi:10.3390/ijpb14020037_

Round 1

Reviewer 1 Report

Aquaculture wastewater irrigation is a viable option (for utilization of wastewater) and has practical relevance, as well. Authors focused on the applicability of wastewater irrigation and salicylic acid elicitation on phenolic compounds production of Mexican oregano (Lippia graveolens) in greenhouse. In my opinion, the topic of the manuscript can be considered as relevant and interesting for the readers. Introduction section summarizes well the background and the relevance of the study. Applied methods are adequate to the sample characteristics and suitable to the main research aims. Manuscript contains valuable results, that are discussed with relevant refernces.

Comments, suggestions:

Please check the typos and stylistic errors (for instance: Aquaculture wastewater is wastewater from aquaculture..’ in line 81-82; ’Steiner solutions is nutrient solution’ in line 84-85; ’ N ha-’ in line 230,… etc)

Please discuss the effects of aquaculture wastewater irrigation and Sa dosage on biomass production rate, as well.

Please give the experimental conditions in a separated sub chapter (see line 75-92).

Please give the standard deviations for data presented in Table 1.

In my opinion, Conclusion is too general; please rephrase to more concrete and give more future outlook of the research, as well.

Reviewer 2 Report

The manuscript is very general research with no innovation data for which can interest the readers. 

1) All the experiments are very simple with all the qualitative results.  2) The results should be based on the quantitation of each of the bioactive compounds present in the Lippia graveolens. 3) Authors need to include the HPLC or LC-MS data for quantifying each of the bioactive compounds.  4) There is not much data collected in the experimental session. Similarly there is not much data to discuss in the results and discussion section.  I would recommend the authors to perform more experiments which can justify their hypothesis. 

Author Response

Present study is an effort to demonstrate the effect of aquaculture wastewater on the stimulation of secondary metabolites in Mexican oregano. Within this context, juvenile fish aquaculture generates wastewater with a supply of nitrogen that can be used by plants. However, the financial resources available at that time allowed us to make determinations on the irrigation water and the plants were to demonstrate the effect on the secondary metabolites, however the HPLC study is in our future objectives.

Round 2

Reviewer 2 Report

Authors did not address any experiment after suggestion. 

Author Response

Dear reviewer, after reading the revision and your suggestion we can´t give more evidence in order to prove our hypothesis because we don´t have any experiments related to our submitted article.

We consider that your contributions to this revision are essential to improve the quality of this work. However, at this moment it is not possible for us to add more information, we apologize for this situation and we will work on it for future projects.